# Long-Chain Branched Bio-Based Poly(butylene dodecanedioate) Copolyester Using Pentaerythritol as Branching Agent: Synthesis, Thermo-Mechanical, and Rheological Properties

**DOI:** 10.3390/polym15153168

**Published:** 2023-07-26

**Authors:** Ruixue Niu, Zhening Zheng, Xuedong Lv, Benqiao He, Sheng Chen, Jiaying Zhang, Yanhong Ji, Yi Liu, Liuchun Zheng

**Affiliations:** 1School of Textile Science and Engineering, Tiangong University, Tianjin 300387, China; 2231010155@tiangong.edu.cn (R.N.); lxd998520@163.com (X.L.); hebenqiao@tjpu.edu.cn (B.H.); 2230010050@tiangong.edu.cn (J.Z.); jiyanhong@tiangong.edu.cn (Y.J.); yiliuchem@whu.edu.cn (Y.L.); 2Hebei Tieke Yichen New Materials Technology Co., Ltd., Shijiazhuang 050000, China; 3Department of Applied Biology and Chemical Technology, Faculty of Science Hong Kong Polytechnic University, Hong Kong 100872, China; zzhening@163.com; 4Technology Innovation Center of Risk Prevention and Control of Refining and Chemical Equipment for State Market Regulation, China Special Equipment Inspection and Research Institute, Beijing 100029, China; 5School of Chemical Engineering and Technology, Xinjiang University, Urumqi 830046, China

**Keywords:** biodegradable, polyester, poly(butylene dodecanedioate), branched, mechanical properties, rheological properties

## Abstract

The introduction of long-chain branched structures into biodegradable polyesters can effectively improve the melt strength and blow-molding properties of polyesters. In this study, pentaerythritol (PER) was used as a branching agent to synthesize branched poly(butylene dodecanedioate) (PBD), and the resulting polymers were characterized by Nuclear Magnetic Resonance Proton Spectra (^1^H NMR) and Fourier Transform Infrared spectroscopy (FT-IR). It was found that the introduction of a small amount of PER (0.25–0.5 mol%) can generate branching and even crosslinking structures. Both impact strength and tensile modulus can be greatly improved by the introduction of a branching agent. With the introduction of 1 mol% PER content in PBD, the notched impact strength of PBD has been increased by 85%, and the tensile modulus has been increased by 206%. Wide-angle X-ray diffraction and differential scanning calorimetry results showed that PER-branched PBDs exhibited improved crystallization ability compared with linear PBDs. Dynamic viscoelastics revealed that shear-thickening behaviors can be found for all branched PBD under low shear rates.

## 1. Introduction

Plastic bags or films made from traditional polymers, such as polyethylene, have become irrecoverable plastic waste after use [1]. This leads to serious environmental problems as these torn bags are discarded, burned, or buried on agricultural land. Agricultural film is another well-known kind of plastic waste. In addition, discarded bags or films will enter waterways and suffocate aquatic life or be eaten by animals, which will have terrible consequences [2,3,4].

Therefore, research on biodegradable polymers has gained increasing interest and developed rapidly in recent years. Although some of these are already industrialized, research into new biodegradable polymers with better thermal and mechanical properties at a lower cost still attracts great interest [5,6]. With regard to biodegradable materials, aliphatic polyesters originated from renewable resources have become a popular topic due to their good biodegradability, acceptable mechanical properties, and low cost [7,8]. Currently, the main bio-based biodegradable polymers include poly(lactic acid), poly(butylene succinate) [9,10], and polyhydroxyalkanoates [11]. They are widely applied as semicrystalline polymers, and some have already been found to be used in the fields of food packaging, medical devices, 3D printing, and agriculture.

What is of particular interest is that bio-based long-chain dicarboxylic acids containing 12 to 20 carbons can be produced on a large scale, and some of them have been industrialized. Polyesters and polyamides have been synthesized from these monomers. Poly(butylene dodecanedioate) (PBD) is a kind of long-chain polyester that can be produced by esterification and polycondensation of 1,4-butanediol (BDO) and dodecanedioic acid (DDCA). The chemical structure and mechanical properties of PBD are similar to those of polyethylene (PE) [12,13,14,15,16]. Compared with traditional plastics like PE and polypropylene, PBD has excellent biodegradability due to the presence of ester bonds in the main chain [17].

For film blowing, especially for ultra-thin films, the melt strength of polymers should be high enough to withstand the stretching of the melt and produce stable, thin, and uniform films [18]. Using multifunctional monomers as branching agents to synthesize long-chain branched polymers is a good choice to change the chain topology and improve the melt strength. Among numerous branching agents, glycerol and pentaerythritol (PER) are the most widely used multifunctional monomers [19]. The combined properties of pentaerythritol-based resin are better than those of glycerol, as the latter has a low reactive secondary hydroxyl group. In addition, dodecanedioic acid has recently been available in industry as a bio-based raw material, making it an attractive monomer for all renewable biodegradable polyester and polyamide [20].

Lu et al. produced the pentaerythritol-branched poly(butylene succinate-*co*-terephthalate) (PBST) with a relative higher Young’s modulus ranging from 120 to 140 MPa and higher melt elasticity or strength due to the branching [21]. Chen et al. developed bio-based unsaturated poly(butylene adipate-*co*-butylene itaconate) (PBABI) using pentaerythritol to adjust the thermo-mechanical properties. The results suggested the PBABI could be a tough elastomer with a Young’s modulus ranging from 65.1–83.8 MPa and exhibit shear-thinning behavior by changing BI contents. Park et al. found that pentaerythritol can enhance the mechanical properties of branched poly(1,4-butylene carbonate-*co*-terephthalate)s (PBCTs). The synthesized PBCTs exhibited both excellent thermal and mechanical properties when the content of pentaerythritol was 0.22 mol% [22]. Branching, as a very useful strategy to improve the mechanical and processing properties of linear polymers, attracts and benefits researchers from many industrialized fields. Santamarıa et al. have reported that poly[ethylene-*co*-(1,4-cyclohexane-terephthalic acid)] (PECT) branched by pentaerythritol (PER) with better flow-induced crystallization because of the favorable effect of branching on the elongation [23].You and Yu et al. reported improved crystallization of PLLA branched by pentaerythritol triacrylate (PETA) and attributed this to increased nucleation density as chain clusters formed in the presence of hydrogen bonding. The synergistic effect of the nucleating agent and plasticizer further improved the crystallization of PLLA [24]. Yi et al. synthesized the star-shaped poly(ε-caprolactone) polyols using trimethylolpropane (or pentaerythritol) as initiator, and the tensile strength and modulus increased gradually with the increase in PER content [25].

PBD is a very new type of biodegradable and biobased polyester, and this is the first time to systematically study the synthesis, structural characterization, and properties of linear PBD and branched PBD. The development and application of biodegradable polymers is one of the most effective ways to solve the problem of plastic pollution. Nevertheless, biodegradable polymers are commonly based on aliphatic linear units with relatively low mechanical properties, melt strength, and heat resistance, which cannot meet the needs of applications. In this work, PER was used as a branching agent to synthesize branched PBD to improve the mechanical and rheological properties of PBD. Macromolecular chain structures of branched PBD were characterized, and the effect of branching on glass transition, mechanical properties, and rheological properties was investigated. It could be found that branching resulted in improved melt strength, impact strength, tensile strength, and tensile modulus. Second, film blowing is a great challenge for most biodegradable polyesters. This work is just to provide an effective strategy to resolve it. PBD with a relatively high branching degree possesses good crystallinity and enhanced melt strength, which are beneficial to film blowing, and the resulting branched PBD has potential applications in sustainable packaging.

## 2. Experimental

### 2.1. Materials

DDCA (99% purity) was provided from Shanghai Aladdin Biochemical Technology Co., Ltd. (Shanghai, China). PER (98% purity), BDO, and stannous octoate (Sn(Oct)_2_) were purchased from J&K Chemical. Phenol, tetrachloroethane, and chloroform were obtained from Aladdin Reagent Co., Ltd. (Shanghai, China). All the raw materials were used without further purification.

### 2.2. Synthesis of PBDx

PBD and PBDx were synthesized by esterification and polycondensation. x in PBDx, which represents the molar percentage of PER based on diacids, was used as a typical sample to describe the synthesis process of branched PBD in detail as follows: DDCA (6 mol), BDO (7.2 mol), and PER (0.03 mol based on diacids) were added to the reaction kettle. Under stirring in a nitrogen atmosphere, the temperature was raised to 160 °C, and then Sn(Oct)_2_ (0.04 wt% on the basis of the total molarity of DDCA) was added as a catalyst. When the theoretical value of water was reached, the temperature was heated to 230 °C and the system pressure was gradually reduced to 20–50 Pa to conduct the polycondensation reaction. When the torque reached a certain value, the polycondensation reaction was stopped.

### 2.3. Characterization

**Structure characterization of PBD and PBDx:** The synthesized branched polyesters were slightly soluble in chloroform, and the molecular weight of the copolymers was difficult to determine as PER leads to branching and even cross-linking of PBD.

Nuclear Magnetic Resonance Proton spectra (^1^H NMR) were measured at 25 °C in CDCl_3_, using TMS as an internal standard, on a 400 MHz Bruker spectrometer (Bruker, AVANCE NEO 400M, Karlsruhe, Germany).

Fourier Transform Infrared spectroscopy (FT-IR) analysis was conducted with the Nicolet 50 Fourier transform infrared spectrometer (Thermo Fisher Scientific, Waltham, MA, USA). The scanning wavenumber ranged from 4000 to 400 cm^−1^. The PBD and PBDx film samples for the test were prepared by hot pressing. Samples of dried polymer films were scanned 32 times to improve the signal-to-noise ratio.

The crystal structures of PBD and PBDx were analyzed by Wide-Angle X-ray Diffraction (WAXD) on a D8 Discover X-ray Diffractometer (German Bruker company, Karlsruhe, Germany). The scanning rate was 2°/min, and the range (2*θ*) was 5–40°.

The intrinsic viscosities of synthesized polymers were tested at 25 °C by an Ubbelohde viscometer (diameter 0.792 nm). Samples of 0.125 g were dissolved in tetrachloroethane/phenol (1/1, *w*/*w*) to achieve a homogeneous solution. The solution concentration was 5 mg/mL. Intrinsic viscosity was obtained from the following equations:$$\eta_{\mathrm{sp}}=\frac{t-t_{0}}{t_{0}}$$
$$\left[\eta \right]=\frac{\sqrt{1+1.4\eta_{\mathrm{sp}}^{-1}}}{0.7C}$$
where η*_sp_* is the specific viscosity, *C* is the concentration, and *t* and *t*_0_ are the flow times of solution and pure solvent, respectively.

**Thermal analysis of PBD and PBDx:** The basic thermodynamic parameters and non-isothermal crystallization of PBD were determined by Differential Scanning Calorimetry (DSC, Netzsch, DSC204F1). Five to ten mg of samples were heated from 0 °C to 130 °C at 20 °C/min, kept there for 5 min to eliminate thermal history, then cooled down to −100 °C at 100 °C/min. Furthermore, it was reheated to 130 °C and kept at this temperature for 5 min. Finally, it was cooled to −60 °C at a cooling/heating rate of 20 °C/min.

**Mechanical properties of PBD and PBDx:** The tensile testing of dumbbell-shaped samples (10.0 × 4.0 × 2.0 cm) was performed using an Instron 1122 tensile tester according to ISO 527. The crosshead speed was 50 mm/min. All tested samples were prepared by injection molding at 175 °C and 800 Bar. At least five samples were tested, and the average value was reported.

**Impact Test:** The impact strength of polymers was tested by Charpy (notch depth 2 mm) using an Instron Dynatup impact tester. The experiment was conducted in accordance with ASTM D 256. During the test, the specimens (80 mm × 10 mm × 4 mm) were placed vertically, and the impactor (projectile) was struck in the center of the specimen. Ten specimens were tested, and the average value was reported.

**Rheological properties of PBD and PBDx:** The dynamic rheological measurements were carried out at 150 °C on a rotational rheometer. The test was performed in the frequency range of 0.01 to 100 rad/s with a fixed strain of 1%. While disc samples with a thickness of 1.1 mm and a diameter of 25 mm were prepared by the HAAKE MiniJet II injection molding machine, prior to testing, the samples were dried in a vacuum drying oven at 70 °C for 8 h to remove the absorbed moisture from the copolymer.

**Determination of the gel fraction of PBD and PBDx:** The samples with a pre-determined weight (*W*_1_) were dissolved in chloroform for 24 h, then stirred and refluxed at 50 °C for 2 days to completely dissolve the uncrosslinked parts. The suspension was centrifuged twice at 10,000 rpm for 30 min; the cross-linked part was collected, washed three times with chloroform, and then vacuum dried at 50 °C to a constant weight (*W*_2_). The gel fraction (*G_f_*) is calculated as follows:(1)$$G_{f}=\frac{W_{2}}{W_{1}}\times 100\%$$

## 3. Result and Discussion

Branched polyesters are usually synthesized by esterification and polycondensation with a small amount of multifunctional comonomers. As shown in Figure 1, PBDx was synthesized in a similar manner to linear PBD, in which PER was used as a branching agent ranging from 0.25 to 1.00 mol% based on diacid. Sn(Oct)_2_ was used as a catalyst for esterification and polycondensation reactions. The reaction conditions and results were summarized in Table 1. The esterification stage involves the generation of tetrahydrofuran (THF) byproducts produced by intramolecular dehydration of BDO. With the increase in PER content, the esterification time and the amount of THF distilled out changed little, suggesting that the introduction of PER did not affect the esterification kinetics. With the increase in PER content, the polycondensation time to a similar viscosity decreased slightly, indicating that a branched structure was formed. When the PER content reached 0.50 mol%, a small amount (0.12%) of gel was generated, and the product could not completely dissolve in chloroform. Each PER molecule has four highly active hydroxyl groups, so it has a high branching ability. The reason for gel formation was that excessive PER caused crosslinking. Therefore, in practical application, the PER content should be lower than 0.8 mol%.

The intrinsic viscosity of the soluble part of PBDx was about 1.20 dL/g, indicating that the addition of 0–0.5 mol% PER did not significantly affect the molecular weight of the soluble part of PBD.

### 3.1. The Chemical Structures and Compositions

The chemical structure of synthesized copolymers was studied by ^1^H NMR. Figure 1 shows the ^1^H NMR spectra of PBD and PBDx based on different PER contents. It can be observed that the spectra of branched PBD were similar to those of unbranched PBD. The peaks at 4.10 ppm and 1.70 ppm were ascribed to the proton ‘a’ and ‘b’ peaks on the BDO residual in the PBD structure; the peaks at 2.25 ppm and 1.60 ppm were stannous proton ‘c’ and ‘d’ peaks on the DDCA residual in the PBD structure; the chemical shifts of proton ‘e’, ‘f’, and ‘g’ peaks from DDCA appeared as multiple peaks at around 1.30 ppm. Since the proton peak on PER and the proton ‘a’ peak from BDO overlapped, the methylene content in PER could not be determined. Therefore, it was impossible to quantitatively calculate the degree of branching. It could be seen from the partially enlarged spectra of Figure 1b that a triplet appeared at 3.68 ppm. The triplet at 3.68 was attributed to methylene hydrogen near the terminal hydroxyl group of the PBD chains. Crosslinking of the PBD chains converts OH groups into ethers, reducing the signal to 3.68 ppm. Probably, the increase in signal is mainly due to the methylene hydrogen near the unreacted OH groups of PER (that partially react). The higher intensity of the terminal hydroxyl group signal could be applied to qualitatively determine that branching degree.

The FT-IR spectra of the PBD and PBDx are displayed in Figure 2. The peaks correlated to the asymmetry and symmetry of C–H stretching absorption were found at 2915 cm^−1^ and 2850 cm^−1^, respectively. The peaks at 1723 and 1266 cm^−1^ corresponded to the C=O bond and the C–O bond on the ester bond, respectively. The intensity of the C–H bond, C=O bond, and C–O bond absorption peaks increased with the increasing PER concentration, which suggested that the PER molecule has been successfully introduced into the main chain of PBD.

The absorption peaks of –OH (3625 cm^−1^ and 3552 cm^−1^) and –H (3462 cm^−1^ and 3422 cm^−1^) were very weak for PBD because of its high molecular weight, which may be due to hydrogen bonds formed by OH groups of PER and OH in the polymer. However, relatively stronger signals were observed in PBD_0.25%_, PBD_0.5%_, PBD_0.75%_, and PBD_1%_, implying more hydroxyls and H–bonds exist in the branched PBD.

In order to characterize the influence of branching degree on the crystal structure of PBD, the polyesters were measured by Wide-angle X-ray diffraction (WAXD), and the results are shown in Figure 3. It can be found that two narrow and sharp diffraction peaks located at 2θ = 21.5° and 24.2° appeared in the pattern, representing the typical (110) and (200) planes of the orthorhombic unit cell, which are consistent with the results reported in the literature [26]. The peak positions of PBD and PE are almost the same [27,28], suggesting they have similar crystal structures. The same diffraction pattern of the PBD and the branched PBD meant that the branching did not change the crystal structure of the PBD. Nevertheless, the peak intensity increased with the degree of branching, indicating that the crystallinity of the copolymer increased with the degree of branching. It may be due to the low molecular weight of every branch, which is beneficial to its crystallization as chain folding and rearranging abilities become easier when the chain is short.

### 3.2. Thermal Properties of PBD and PBDx

The thermal properties of PBD and PBDx were characterized by DSC. The non-isothermal crystallization and melting processes of the obtained polymers were characterized and presented in Figure 4. The relevant thermal performance parameters are summarized in Table 2. All samples showed evident crystallization peaks during the cooling process. When the 0.25 mol% PER was introduced, the crystallization temperature (*T*_c_) decreased from 52.2 °C of PBD to 45.9 °C, and the melting temperature (*T*_m_) of PBD increased from 74.5 °C of PBD to 77.0 °C. This was because the branching structures of PBDx act as nucleators and increase the lamella thickness [29]. Crystallization enthalpy (Δ*H*_c_) increased from 87.1 to 93.0 J/g, and melting enthalpy (Δ*H*_m_) increased from 81.7 to 93.5 J/g as the PER content increased. These results are due to the increased crystallinity of branched PBDx because branching structures generated in PBDx can act as nucleators [25]. The increased width-in-half of the crystallization and melting peaks indicates a decreasing crystallization rate with increasing PER content. Therefore, it can be concluded from the DSC results that PER-branching has improved *T*_m_ and *X*_c_ and enhanced crystallization.

### 3.3. Mechanical Properties of PBD and PBDx

The influence of PER content on the mechanical properties of PBDx is shown in Figure 5. The notched impact strengths of the PBD and PBDx are shown in Figure 5a. The branching formation resulted in a high impact strength of up to 444 J/m, suggesting that the material displayed super-toughness.

The tensile curves, the tensile modulus, elongation at yield, and elongation at break of linear PBDs and branching PBDs are displayed in Figure 5. It can be seen from Figure 5 that the tensile strength and Young’s modulus increased by 2.6 times and 3.1 times with the increase in PER content, respectively. The elongation at break reduced significantly because of the formation of crosslinked chains, and tensile strength increased significantly with the increase in FRE content, which indicates that the copolymer material can be hard plastics or elastomers, depending on the content of pentaerythritol.

### 3.4. Dynamic Viscoelastic Properties of PBD and PBDx

The topological structure of a polymer can directly affect its dynamic rheological behavior. In general, branched polymers exhibit different viscoelasticity behaviors from linear polymers, such as higher melt strength and longer relaxation times. Therefore, the dynamic rheological properties of linear PBD and branched PBDx were studied at 150 °C (that is, the general processing temperature of PBD), and the influence of branching degree on the storage modulus (G′) and loss modulus (G″) was explored.

It can be observed from Figure 6 that when the loss modulus (G″) value was the same, the storage modulus (G′) value increased regularly with increasing PER content. The results indicate that a higher degree of branching leads to an increase in the melt strength of the copolymer. When the loss modulus (G″) was 10^4^ Pa, the storage modulus (G′) of branched PBD_0.5%_ was six times higher than that of linear PBD. In short, a small amount of PER incorporation was sufficient to improve the melt elasticity or strength of PBD, which was very important for film blowing processing.

The rheological properties of PBD and branched PBD with a continuous plastic matrix are the key factors affecting the melt processability, recyclability, and production efficiency of polyesters. A modular compact rheometer was applied to determine the rheological behavior of PBD and PBDx; the results are shown in Figure 7. The apparently constant viscosities of the PBD decrease with increasing shear rate; this indicates the initial increase in apparently constant viscosities was temporary and caused by the polymer that resists the initial movement. The viscosity of the branched PBD first increases evidently with an increase in shear rate, achieves a maximum value, and then decreases with a further increase in shear rate, implying a transition from an expanding fluid to a pseudoplastic fluid. The viscosities of the branched PBDs were much higher than those of PBD at low shear rates, implying the formation of a strong entanglement between the PBD molecular chains. Generally, the microstructure of the polymer determines its rheological properties. The crosslinked polymer melt was not easily deformed and could maintain its morphology in the melt even at high shear rates, resulting in higher viscosities of PBDx than those of PBD. However, under high shear rates, the breakdown of the crosslinked networks formed by PER caused an evident decrease in viscosity [30]. Furthermore, the viscosity of PBDx increased with the increase in PER content. This result was mainly attributed to an increase in the number and density of branches, resulting in a stronger cross-linked network. Notably, the viscosities of branched PBDs were below 10^3^ Pa·s under high shear rates, suggesting that they have good processibility.

## 4. Conclusions

Long-chain branched PBD were synthesized from dodecanedioic acid and 1,4-butanediol by melt polycondensation with 0.25–1 mol% pentaerythritol as branching agent. The prepared branched polyesters have good mechanical properties, elasticity, and processability. More importantly, the branching PBDs obtained have better crystallization abilities than the linear PBDs. Furthermore, a small amount of PER (0.25–0.5 mol%) is sufficient to improve the toughness, modulus, and melt strength of PBD, thus improving its film-blowing properties. PBD with a relatively high branching degree possesses good crystallinity and enhanced melt elasticity or strength, which have potential applications in sustainable packaging.

## Data Availability

Not applicable.

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
