# Peer review of "Long-Chain Branched Bio-Based Poly(butylene dodecanedioate) Copolyester Using Pentaerythritol as Branching Agent: Synthesis, Thermo-Mechanical, and Rheological Properties"

_polymers, 2023, doi:10.3390/polym15153168_

Round 1

Reviewer 1 Report

The aim of the study was to use pentaerythritol as a branching agent for the synthesis of branched poly(butylene dodecanedioate) in order to improve the mechanical and rheological properties of the polymer. There is no clearly presented novelty for the Readers. Due to the fact that there is a lot of research in this area (use of pentaerythritol as branching agent), please provide a clear aim and explicit novelty of the work. Punctuation should be checked because the text contains many minor errors (e.g. capital letters in the middle of a sentence, space after "poly", in -co-: c italic and o not anymore), etc. The manuscript contains a lot of mistakes (some highlighted below), which should be corrected.

Line 46: “polylactic acid”, “poly(hydroxyalkanoates)”; Please use the IUPAC nomenclature. The names of polymers whose monomers consist of two words or more are written with parentheses and vice versa without parentheses when monomer consist of one word. Please correct throughout the text.

Line 55-56: “Compared to the traditional plastic PE, PBD has excellent biodegradability because of the existence of ester bonds in the PBD main chain” English should be checked as the sentences are ambiguous. Please correct throughout the text.

Table 1, 2: Variables should be explained below the table.

Page 8: According SI variables should be in italics. Please correct throughout the text.

Author Response

Dear editors and referees,

Thank you very much for your careful and thoughtful work on our manuscript ID Manuscript polymers-2478372 entitled “Long chain branched bio-based poly (butylene dodecanedioate) copolyester using pentaerythritol as branching agent: Synthesis, thermos-mechanical and rheological properties”.

We appreciate the reviewers’ professional comments and suggestions, which are helpful to improve the quality of our manuscript. We have carefully considered the comments and revised our manuscript accordingly. Our point-to-point responses to the reviewers’ comments are also listed in the following response part. The revised manuscript has covered all the reviewers’ comments. The submitted revision includes the response letter and the revised manuscript. We hope that the revised manuscript is suitable for publication on Polymers.

We would like to appreciate your consideration for our manuscript. If you have any questions on the manuscript, please feel free to contact us.

Yours sincerely,

Liuchun Zheng

Response to reviewers

The aim of the study was to use pentaerythritol as a branching agent for the synthesis of branched poly(butylene dodecanedioate) in order to improve the mechanical and rheological properties of the polymer. There is no clearly presented novelty for the Readers. Due to the fact that there is a lot of research in this area (use of pentaerythritol as branching agent), please provide a clear aim and explicit novelty of the work. Punctuation should be checked because the text contains many minor errors (e.g. capital letters in the middle of a sentence, space after "poly", in -co-: c italic and o not anymore), etc. The manuscript contains a lot of mistakes (some highlighted below), which should be corrected.

Line 46: “polylactic acid”, “poly(hydroxyalkanoates)”; Please use the IUPAC nomenclature. The names of polymers whose monomers consist of two words or more are written with parentheses and vice versa without parentheses when monomer consist of one word. Please correct throughout the text.

Response: “polylactic acid” have been revised “poly(lactic acid)” poly(hydroxyalkanoates) has been corrected as polyhydroxyalkanoates.

Line 55-56: “Compared to the traditional plastic PE, PBD has excellent biodegradability because of the existence of ester bonds in the PBD main chain” English should be checked as the sentences are ambiguous. Please correct throughout the text.

Response: Compared with traditional plastic polyethylene and polypropylene, PBD has excellent biodegradability due to the presence of ester bonds in the main chain.

Table 1, 2: Variables should be explained below the table.

Sample

aφPER (%)

bt (min)

c[η] (dL/g)

eGf (%)

PBD

0

65

1.05

0

PBD0.25%

0.25

60

1.20

0.12

PBD0.5%

0.50

50

1.26

1.68

PBD0.75%

0.75

55

1.15

3.03

PBD1%

1.00

60

-d

66.13

a Molar percentage of PER based on diacids; b Melt polycondensation time. c Intrinsic viscosity measured using tetrachloro-ethane/phenol (1/1, w/w) as solvent; d Samples with gel observed when dissolved in chloroform; e Gel fraction of samples.

Sample

Tc ()

Tm()

ΔHc (J/g)

ΔHm (J/g)

PBD

54.4

74.5

87.1

81.7

PBD0.25%

52.2

77.2

90.2

82.0

PBD0.5%

47.2

79.5

88.5

82.1

PBD0.75%

46.8

80.6

94.1

91.4

PBD1%

45.9

82.3

93.0

93.5

Tc (℃) crystallization temperature;    Tm (℃) melting temperature;ΔHc(J/g) enthalpy of crystallization;  ΔHm(J/g) melting enthalpy of crystallization.

Page 8: According SI variables should be in italics. Please correct throughout the text.

Response: We have changed the variable to italic. Such as ‘’branched PBDx’’ have been revised’’PBDx’’. ‘’PBD0.25%, PBD0.5%, PBD0.75%, and PBD1%, implying more hydroxyls and H-bonds exist in the branched PBD.’’ have been corrected ‘PBD0.25%, PBD0.5%, PBD0.75%, and PBD1%, implying more hydroxyls and H-bonds exist in the branched PBD’.

Reviewer 2 Report

Dear Authors,

I reviewed your intriguing manuscript, which presents the production and characterization of bio-based poly(butylene dodecanedioate) copolyesters synthesized from dodecanedioic acid and 1,4-butanediol by melt polycondensation, branched with pentaerythritol. It was shown that small amounts of pentaerythritol result in branching and crosslinking of the polyester chains, which in turn improves toughness (impact strength and tensile modulus), and improves crystallization behavior. The generated materials are suggested as novel polymers for packaging applications.

The work is topical and scientifically sound, the experiments are well planned, and the conclusions are logical and substantiated by the elaborated experimental data. State of the art methodology was used for material characterization: 1H-NMR, FTIR, DSC, rheometers, etc.

Some aspects, however, need to be addressed in order to further improve your submission:

a)       Title: “thermos-mechanical“: should be „thermo-mechanical”; line 71: “thermal-mechanical”: should be „thermo-mechanical”

b)      Entire manuscript needs to be revised by a person skilled in scientific English writing; there are many grammar shortcomings, awkward sentence constructions, etc. just some examples: line 19: “resulted”: “resulting”; line 21: “can generates”: “can generate”; line 45: “originated from”: “originating from”; line 91: “Stannous”: “stannous”; line 130: “were conducted accordance with”: “were conducted in accordance with”; line 138: “Determination the gel fraction”: “Determination of the gel fraction”; line 138: “with predeterminate weight”: “with predeterminated weight”; “Kinetics”: “kinetics”; line 160: “cause”: “causing”; line 226: “These result from the fact”: “These results are due to the fact”; line 252: “of polymer”: “of a polymer”; line 272: “rate, indicates a transition”: “rate; this indicates a transition”; line 274: “with increase with shear rate”: “with increase of shear rate”

c)       Keywords: I am surprised that you use “Biodegradable”, but “Bio-based” in the title. What is correct? Both?

d)      Did you perform any biodegradability tests with the new materials in order to assess the impact of branching and crosslinking by pentaerythritol in biodegradability?

e)      Line 45-46: Check the references! polylactic acid (should be poly(lactic acid)) has no references, while in your list of references, ref. [11] refers to poly(lactic acid); ref. [11] does not fit to polyhydroxyalkanoate (PHA), where it is placed. Here, I suggest to add a new reference to a recent review on PHA, e.g., Mukherjee, A., & Koller, M. (2022). Polyhydroxyalkanoate (PHA) Biopolyesters-Emerging and Major Products of Industrial Biotechnology. The EuroBiotech Journal6(2), 49-60. Check references in entire text!

f)        Line 63: what is “unstable carbon”? do you mean “unstable carbon-carbon bonds”?

g)       Line 67: “et al”: should be “et al.”; the same goes for line 69 (“et.al.”)

h)      Line 72: “young´s”: “Young´s”

i)        Line 98: “nitrogen protection”: do you mean “under nitrogen atmosphere”?

j)        Line 135: what is a “Austria Antonpa Co., Ltd., MCR302, Austria”? do you mean the rheometer MCR 302, Anton Paar, Austria?

k)       Line 140: “10000 rpm”: express as “rcf” (multiple of g), or add type of centrifuge used in order to make the process reproducible for peers.

l)        Fig. 5 and its interpretation: did you also determine glass transition temperature (Tg)? If there is no, this should be discussed and interpreted.

m)    Line 226: “Therefore, the materials can be hard plastics or soft elastomer”: not clear. Are they hard plastics or elastomers? Does this depend on addition of pentaerythritol? Please, rephrase!

n)      Line 246: “Proper content”: for what?

o)      Fig. 6: why are values in Fig. 6d shown as points, while there are bars presented in Fig. 6a, 6b and 6c? Why are the bars in 6a and 6b in pink, but in blue in 6c? This is confusing for the readers!

p)      Line 257: remove short sentence “As shown in Figure 7.”; it makes no sense.

q)      List of references: check page range of reference [6] (Tsui et al., 2013)

see comments to authors; I listed several examples where English needs to be improved

Author Response

Dear editors and referees,

Thank you very much for your careful and thoughtful work on our manuscript ID Manuscript polymers-2478372 entitled “Long chain branched bio-based poly (butylene dodecanedioate) copolyester using pentaerythritol as branching agent: Synthesis, thermos-mechanical and rheological properties”.

We appreciate the reviewers’ professional comments and suggestions, which are helpful to improve the quality of our manuscript. We have carefully considered the comments and revised our manuscript accordingly. Our point-to-point responses to the reviewers’ comments are also listed in the following response part. The revised manuscript has covered all the reviewers’ comments. The submitted revision includes the response letter and the revised manuscript. We hope that the revised manuscript is suitable for publication on Polymers.

We would like to appreciate your consideration for our manuscript. If you have any questions on the manuscript, please feel free to contact us.

Yours sincerely,

Liuchun Zheng

Response to Reviewer Comments

I reviewed your intriguing manuscript, which presents the production and characterization of bio-based poly(butylene dodecanedioate) copolyesters synthesized from dodecanedioic acid and 1,4-butanediol by melt polycondensation, branched with pentaerythritol. It was shown that small amounts of pentaerythritol result in branching and crosslinking of the polyester chains, which in turn improves toughness (impact strength and tensile modulus), and improves crystallization behavior. The generated materials are suggested as novel polymers for packaging applications.

The work is topical and scientifically sound, the experiments are well planned, and the conclusions are logical and substantiated by the elaborated experimental data. State of the art methodology was used for material characterization: 1H-NMR, FTIR, DSC, rheometers, etc.

Some aspects, however, need to be addressed in order to further improve your submission:

Response: Thanks very much for your comments, which are very helpful to improve the quality of this article.

  1. a) Title: “thermos-mechanical“: should be „thermo-mechanical”; line 71: “thermal-mechanical”: should be „thermo-mechanical”

thermos-mechanical

Response: Title: “thermos-mechanical“ has been corrected as “thermo-mechanical” and line 71“thermal-mechanical”has been revised as“thermo-mechanical”

  1. b) Entire manuscript needs to be revised by a person skilled in scientific English writing; there are many grammar shortcomings, awkward sentence constructions, etc. just some examples: line 19: “resulted”: “resulting”; line 21: “can generates”: “can generate”; line 45: “originated from”: “originating from”; line 91: “Stannous”: “stannous”; line 130: “were conducted accordance with”: “were conducted in accordance with”; line 138: “Determination the gel fraction”: “Determination of the gel fraction”; line 138: “with predeterminate weight”: “with predeterminated weight”; “Kinetics”: “kinetics”; line 160: “cause”: “causing”; line 226: “These result from the fact”: “These results are due to the fact”; line 252: “of polymer”: “of a polymer”; line 272: “rate, indicates a transition”: “rate; this indicates a transition”; line 274: “with increase with shear rate”: “with increase of shear rate”

Response: Thanks very much for your comments, which are very helpful to improve the quality of this article. We have checked the whole manuscript very carefully, just some examples: line 19: “resulted” has been revised “resulting”; line 21: “can generates” has been revised “can generate”; line 45: “originated from” has been revised “originating from”; line 91: “Stannous” has been corrected as “stannous”; line 130: “were conducted accordance with” has been corrected as “were conducted in accordance with”; line 138: “Determination the gel fraction” has been corrected as “Determination of the gel fraction”; line 138: “with predeterminate weight” has been revised as “with predeterminated weight”; “Kinetics” has been revised as “kinetics”; line 160: “cause” has been replaced with “causing”; line 226: “These result from the fact” has been corrected as “These results are due to the fact”; line 252: “of polymer” has been revised as “of a polymer”; line 272: “rate, indicates a transition” has been replaced with “rate; this indicates a transition”; line 274: “with increase with shear rate” has been revised as “with increase of shear rate” as suggested.

  1. c) Keywords: I am surprised that you use “Biodegradable”, but “Bio-based” in the title. What is correct? Both?

Response: Poly(butylene dodecanedioate) can be biodegradable and it is partially bio-based as 1,12-dodecanedicarboxylic acid can be originated from biomass.

  1. d) Did you perform any biodegradability tests with the new materials in order to assess the impact of branching and crosslinking by pentaerythritol in biodegradability?

Response: We are deeply grateful for this professional suggestion; the degradation of polymers in soil and similar microbial environments is known to be critical, but there is no way to conduct this experiment as it needs 3 months or longer time, and there is only 10 days for us to revise and return the manuscript. We will study the degradation behavior in soil degradation and similar microbial environments in a subsequent study. In fact, we have studied the biodegradation of PBD by enzyme and the results show that the biodegradation rate is very fast. PBD film can loss 80 wt% weight in four days, which is much faster than that of PBS.

  1. e) Line 45-46: Check the references! polylactic acid (should be poly(lactic acid)) has no references, while in your list of references, ref. [11] refers to poly(lactic acid); ref. [11] does not fit to polyhydroxyalkanoate (PHA), where it is placed. Here, I suggest to add a new reference to a recent review on PHA, e.g., Mukherjee, A., & Koller, M. (2022). Polyhydroxyalkanoate (PHA) Biopolyesters-Emerging and Major Products of Industrial Biotechnology. The EuroBiotech Journal, 6(2), 49-60. Check references in entire text!

Response: According to the views of the reviewer, we have checked the reference [11] and review more recent works to convince readers that such investigation can also benefit other polyester modifications.

  1. Mukherjee, A., Koller,M.(2022).Polyhydroxyalkanoate (PHA) Biopolyesters - Emerging and Major Products of Industrial Biotechnology. The EuroBiotech Journal,6(2) 49-60.
  2. f) Line 63: what is “unstable carbon”? do you mean “unstable carbon-carbon bonds”?

Response: “unstable carbon” is the quaternary carbon originated from pentaerythritol.

  1. g) Line 67: “et al”: should be “et al.”; the same goes for line 69 (“et.al.”)

Response: ‘’et al’’ and “et.al.” have been revised “et al”.

  1. h) Line 72: “young´s”: “Young´s”

Response: “young´s” has corrected “Young´s”.

  1. i) Line 98: “nitrogen protection”: do you mean “under nitrogen atmosphere”?

Response: “nitrogen protection” has corrected “under nitrogen atmosphere”.

  1. j) Line 135: what is a “Austria Antonpa Co., Ltd., MCR302, Austria”? do you mean the rheometer MCR 302, Anton Paar, Austria?

Response: “Austria Antonpa Co., Ltd., MCR302, Austria” has been revised ‘’rheometer MCR 302, Anton Paar, Austria’’.

  1. k) Line 140: “10000 rpm”: express as “rcf” (multiple of g), or add type of centrifuge used in order to make the process reproducible for peers.

Response: we used VL-8F centrifuge (Hunan Ying Hong Technology Co. China changsha).

  1. l) 5 and its interpretation: did you also determine glass transition temperature (Tg)? If there is no, this should be discussed and interpreted.

Response: The glass transition temperature of the copolymer cannot be observed in the DSC curve as shown in the Fig. 5 as the glass transition temperature should be well below 0 ℃, and the test temperature range is 0-130 ℃.

  1. m) Line 226: “Therefore, the materials can be hard plastics or soft elastomer”: not clear. Are they hard plastics or elastomers? Does this depend on addition of pentaerythritol? Please, rephrase!

Response: Yes, it depends on the content of pentaerythritol. As shown in Figure 6(d), the copolymer material elongation at break decreases, while tensile strength increases, significantly with the increase of FRE content, which indicates that the copolymer material can be hard plastics or elastomers, depending on the content of pentaerythritol.

  1. n) Line 246: “Proper content”: for what?

Response: Suitable amount of PER is good for regulating the processing properties, mechanical properties and thermal stability of PBD.

  1. o) 6: why are values in Fig. 6d shown as points, while there are bars presented in Fig. 6a, 6b and 6c? Why are the bars in 6a and 6b in pink, but in blue in 6c? This is confusing for the readers!

Response: In Figure 6d, the dotted Figure 6d has been revised as solid line to make it more-easy to understand. The colors of the bars have all been revised to purple.

  1. p) Line 257: remove short sentence “As shown in Figure 7.”; it makes no sense.

Response: We have removed “As shown in Figure 7.” as suggested.

  1. q) List of references: check page range of reference [6] (Tsui et al., 2013)

Response: We have listed the reference [6] (Tsui et al., 2013); please see the following or the reference part.

[6] Tsui, A.; Wright, Z.C.; Frank, C.W. Biodegradable polyesters from renewable resources. Annu. Rev. Chem. Biomol. Eng. 2013, 4, 143-170.

Reviewer 3 Report

Please see file attached

Author Response

Dear editors and referees,

Thank you very much for your careful and thoughtful work on our manuscript ID Manuscript polymers-2478372 entitled “Long chain branched bio-based poly (butylene dodecanedioate) copolyester using pentaerythritol as branching agent: Synthesis, thermos-mechanical and rheological properties”.

We appreciate the reviewers’ professional comments and suggestions, which are helpful to improve the quality of our manuscript. We have carefully considered the comments and revised our manuscript accordingly. Our point-to-point responses to the reviewers’ comments are also listed in the following response part. The revised manuscript has covered all the reviewers’ comments. The submitted revision includes the response letter and the revised manuscript. We hope that the revised manuscript is suitable for publication on Polymers.

We would like to appreciate your consideration for our manuscript. If you have any questions on the manuscript, please feel free to contact us.

Yours sincerely,

Liuchun Zheng

Response to Reviewer Comments

The manuscript is focused on the study and characterization of branched PBD as innovative biodegradable polymer. The study of new materials and the methods to increase their performance makes this article undoubtedly innovative. Description of the polymerization must be improved because the experiment should be repeatable by any laboratory that read this manuscript. Reading the paper I had the impression that the materials were characterized by numerous different technique (I recognize a huge effort) but few of them were in the expertise of the writers. Despite this, I think that a complete characterization of these innovative materials deserves publication and interest from the scientific community.

Line 54: “The chemical structure of PBD is similar to PE” A carbon chain full of ester groups cannot be compared to a fully carbon chain (with no oxygen). Please be clearer or change this affirmation.

Response: The chemical structure and mechanical properties of PBD are similar to polyethylene (PE) [12-16]. Compared with traditional plastic PE, PBD has excellent biodegradability due to the presence of ester bonds.

Line 71: “PBABI could be rigid plastics or elastomers with young’s modulus ranging from 65.1-83.8 MPa” A plastic with these range of modulus cannot be described as rigid plastic.

Response: The results suggested the PBABI could be tough elastomer with Young’s modulus ranging from 65.1-83.8 MPa and exhibited shear-thinning behavior by changing BI contents.

I suggest to move Scheme 1 in the 2.2 section.

Response: We have moved Scheme 1 in the 2.2 section as suggested.

Line 95 it is the first time that I encounter the subscript 0.5% of the word PBD and it should be defined here. I suggest to move the sentence “PBD0,5% was used as a typical sample to describe the synthesis process of branched PBD” at the end of the paragraph (after “x in PBDx represents the molar percentage of PER based on diacids”)

Response: We have corrected it as suggested.
Line 97 Molar quantity of reagents were chosen on the basis of literature or previous experiments? Eventually cite these articles or explain the choice of these molar ratios.

Response: Molar quantity of reagents were chosen on the basis of previous experiments.
Line 98 nitrogen protection is not fully correct because nitrogen did not protect anything (it is inert). Is suggest “under stirring and nitrogen as inert gas”

Response: “nitrogen protection” has been corrected as “under nitrogen atmosphere” as suggested.
Line 99 The quantity of catalyst was decided on the basis of literature?

Response: The catalyst dosage is based on our previous studies [1] and Sn(Oct)2 amount is 0.04 wt% on the basis total molar of DDCA.

[1] Lv, Xuedong, et al. "Synthesis of Biodegradable Polyester–Polyether with Enhanced Hydrophilicity, Thermal Stability, Toughness, and Degradation Rate." Polymers 14.22 (2022): 4895.
Line 99 Define the amount of theorical water and how you evaluate it (the water was produced under a close flask, how do you measure its volume/weight?) If water was present how is possible to maintain 160 °C? The water was extracted during the reaction? Can you add a scheme of the reaction kettle?

Response:  Amount of theorical water was determined by weight. Under the reaction temperature of 160 °C, generated amount of side product water at certain time is low and tends to evaluate in the form of a gas and be collected. Thus, the water will not retain in the reactor and affect the reaction temperature. The scheme of the reaction kettle is shown as following:

Line 100: Define on which basis you choose reaction temperature. The vacuum was used for remove water?

Response: We choose reaction temperature based on the reaction. Usually, the reaction rate should be fast enough and side reaction should be low under an optimal temperature. The vacuum was used to remove butanediol and small amount of water.

Line 101 Define the value of torque at which the reaction end. Also define the instrument for the measurement of the torque (I suppose is a function of the mechanical stirrer)

Response: The torque at the reaction end is 16 N.m at 50 rpm. The mechanical stirrer has been equipped with a torque sensor.
Line 102 The polycondensation reaction was stopped opening the kettle or adding some additive?

Response: The polycondensation reaction was stopped without adding any additive.
Line 104 purification of the obtained polymers were not performed?

Response: The extent of the reaction was perfect and no further purification of the sample was required. This is common for the industrialization of polyesters, such as PET and PBT.
Line 115 define pressure and temperature of hot press and also thickness of obtained films.

Response: All tested samples were prepared by injection molding at 125 oC and 800 Bar. Film samples with thickness of 80 μm were obtained by pressing the polyester material on a hot-press under the identical pressure and temperature.
Line 119 I suppose that at 25 the polymer was solid. How can you measure its viscosity? If you use a solvent define it and also the concentration of polymer solution.

Response: The process to determine the viscosity has been supplied; please see “2.3. Characterization” or the following: 0.125 g samples were dissolved in tetrachlor-oethane/phenol (1/1, w/w) to achieve a homogeneous solution. The solution concentration was 5 mg/mL.
Line 126 standard samples according to ISO 527 and Iso 180? If not define the ISO of specimen dimension. Specimen for tensile and impact required different amount of polymer, is suggest to remove the specification of 15g for the injection molding machine.

Response: The tensile testing of dumbbell-shaped samples (10.0x4.0 x2.0 cm) was performed using an Instron1122 tensile tester according to ISO 527. The specification of 15g for the injection molding machine” has been deleted as suggested.
Line 130 define speed test and initial grip separation

Response: The cross-head speed was 50 mm/min.
Line 131 Define length of the notch and the joules of the pendulum. Mechanical specimens were conditioned before the tests?

Response: The impact strength of polyester was tested by through Charpy (notch depth 2 mm) using Instron Dynatup impact tester. The experiments were carried out by following in accordance with ASTM D256-2010. During the test, the specimens (80 mm × 10 mm ×4 mm) were placed freely and the impactor (The impactor (projectile) is struck in the center of the specimen.
Table 1 “t(min)” was not defined under the table. If is related to the reaction time, how you justify the higher reaction time for PBD? As you report “When the torque reached a certain value, the polycondensation reaction was stopped”. PBD1% has a very high gel fraction and theoretically the torque “limit value” should be reached earlier than PBD0.5% but the reaction time of PBD1% is higher than PBD0.5%.

Response: bt is the PBD condensation reaction time. Polycondensation time changes little with the PER content as the polycondensation time is also closely related with the vacuum degree.
Line 180 triplet at 3.68 was attributed at methylene hydrogen near terminal hydroxyl group of the PBD chains. Crosslinking of PBD chain converts OH groups into ethers reducing the signal at 3.68ppm. Probably the increase in signal is mainly due to the methylene hydrogen near unreacted OH groups of PER (that partially react).

Response: We have added the comments about 1H NMR date as suggested. Triplet at 3.68 was attributed at methylene hydrogen connected to terminal hydroxyl group of the PBD chains. Crosslinking of PBD chain does not reduce the signal at 3.68 ppm, but enhance it as more terminal groups exist due to their more branches.

Line 196 stretching of OH band (alcol) is very wide and generally can be found approximatively in the range 3200-3550 cm-1. I’ve never heard the -H signal at 3462 and 3422 cm-1. IR bands were relative to bond movements (mainly stretching and bending) and writing -H it was equal to C-H (that was at 2915 and 2850). Please better explain what is -H. Regarding -OH band I saw from your spectra a slightly increase of the wide-OH band increasing the PER concentration. These results could be in accordance with my previous theory that H-NMR signals at 3.68 were correlated to unreacted OH groups of PER.

Response: -H signals at 3462 and 3422 cm-1 may be due to hydrogen bonds formed by OH groups of PER and OH in the polymer.
Figure 5 caption. I suggest to change in “DSC a) cooling and b) second heating curves of PBD and PBDx”
Response: We have corrected as DSC ‘’a) cooling and b) second heating curves of PBD and PBDx’’ as suggested.

Line 244 tensile strength was not reported in figure 6

Response: In Figure 6(e), the tensile strength of the polymer increases evidently with the increase of FRE content.
Line 247 Proper content depends from the final application. For 1GPa of elastic modulus a 1% of PER content was required. I suggest to remove this phrase.

Response: We have removed this phrase as suggested.

Line 274 it is only a pseudoplastic fluid behaviour (as most of the polymers). No transition was observed. At low shear rate the reduction of viscosity was lower. Initial increase was temporary caused by the polymer that resist to the initial movement.

Response: We totally agree with your viewpoint and revised it correspondently. Please see Page 9 or the following: Initial increase in apparently constant viscosities was temporary caused by the polymerthat resist to the initial movement.

Line 284 please explained the reason of this particular behavior (breakdown of PBD 1%).

Response: The melt viscosity of the branched PBD was below 103 Pa.s at high shear rates (10-100 s-1), and the PBD1% behaved as an unstable fluid, which was mainly due to a significant reduction in viscosity caused by the broken of the branching entanglement and crossed structure in the PBD1% under strong shear force.
Figure 8 in the previous caption you write PBDx and her PBDs. Choose one and use it for all the caption.

Response: We have corrected the caption as‘ Shear viscosity as a function of shear rate for PBD and PBDx’’.
Line 300 “has broad application prospect in the field of sustainable packaging

Response: We have revised it as “which has potential applications in sustainable packaging.“

Reviewer 4 Report

The authors present their work on the synthesis of branched, long alkyl chain poly(butylene dodecanedioate) copolyester.  The authors used pentaerythritol as a branching agent at low concentration to minimize gelation of the final product.  The different polymers made showed changes in crystallization behavior and mechanical properties (tensile properties, shear modulus, and viscosity) with increasing concentration of pentaerythritol.  Developing new polyesters is significant in the expansion of biodegradable commercial materials.

While there are no specific issues with the manuscript, it is difficult to make any concrete conclusions from the data.  The nature of the crystallization behavior and mechanical properties depends highly on the molecular structure (branching, length of branches (possibility of star-like), overall molecular weight and PDI).  The reviewer understands that these types of characterizations are difficult; however, it is critical to have some understanding of molecular structure to make conclusions about the physical properties of the materials.  This is particularly true with such small changes to the amount of pentaerythritol added.  It is recommended that the manuscript not be published until additional characterization of the molecular structure, even if a bit crude, is provided.

A couple other notes: The x-ray diffraction peaks seem slightly shifted for the copolymer, suggesting a slightly different structure compared to pure PBD.  Additionally, the DSC traces also seem to show multiple melting/crystallization peaks that could indicate mixtures in structure or in fractionated crystallization.  The FTIR results regarding -OH groups remaining in the polymer is likely inconclusive at such a high signal-to-noise ratio in the range of 3400 - 3800 cm-1.

Very minor editing/proofreading needed throughout

Author Response

Dear editors and referees,

Thank you very much for your careful and thoughtful work on our manuscript ID Manuscript polymers-2478372 entitled “Long chain branched bio-based poly (butylene dodecanedioate) copolyester using pentaerythritol as branching agent: Synthesis, thermos-mechanical and rheological properties”.

We appreciate the reviewers’ professional comments and suggestions, which are helpful to improve the quality of our manuscript. We have carefully considered the comments and revised our manuscript accordingly. Our point-to-point responses to the reviewers’ comments are also listed in the following response part. The revised manuscript has covered all the reviewers’ comments. The submitted revision includes the response letter and the revised manuscript. We hope that the revised manuscript is suitable for publication on Polymers.

We would like to appreciate your consideration for our manuscript. If you have any questions on the manuscript, please feel free to contact us.

Yours sincerely,

Liuchun Zheng

Response to Reviewer Comments

The authors present their work on the synthesis of branched, long alkyl chain poly(butylene dodecanedioate) copolyester.  The authors used pentaerythritol as a branching agent at low concentration to minimize gelation of the final product.  The different polymers made showed changes in crystallization behavior and mechanical properties (tensile properties, shear modulus, and viscosity) with increasing concentration of pentaerythritol.  Developing new polyesters is significant in the expansion of biodegradable commercial materials.

While there are no specific issues with the manuscript, it is difficult to make any concrete conclusions from the data.  The nature of the crystallization behavior and mechanical properties depends highly on the molecular structure (branching, length of branches (possibility of star-like), overall molecular weight and PDI).  The reviewer understands that these types of characterizations are difficult; however, it is critical to have some understanding of molecular structure to make conclusions about the physical properties of the materials.  This is particularly true with such small changes to the amount of pentaerythritol added.  It is recommended that the manuscript not be published until additional characterization of the molecular structure, even if a bit crude, is provided.

Response: In fact, this branching work is very novel and the results are very interesting and useful. I believe it will attract and benefit researcher from many industrialized filed. First, PBD is a very new type of biodegradable and biobased polyester. Second, film blowing is a great challenge for most biodegradable polyester. This work is just to provide some effective strategy to resolve it. As far as more characterization of the molecular structure is regarded, the terminal date to return the paper is July 7, 2023, only two days left for us to revise our paper. Therefore, we have no time to book and conduct additional characterization.

A couple other notes: The x-ray diffraction peaks seem slightly shifted for the copolymer, suggesting a slightly different structure compared to pure PBD.  Additionally, the DSC traces also seem to show multiple melting/crystallization peaks that could indicate mixtures in structure or in fractionated crystallization.  The FTIR results regarding -OH groups remaining in the polymer is likely inconclusive at such a high signal-to-noise ratio in the range of 3400 - 3800 cm-1.

Response: I agree with you that PBDx has a slightly different structure compared to pure PBD. But DSC traces does not show multiple melting/crystallization peaks as only one melting/crystallization peak is observed.

Comments on the Quality of English Language Very minor editing/proofreading needed throughout

Response: We have checked and polished the English Language thoroughly and very carefully. Just some examples: line 19: “resulted” has been revised “resulting”; line 21: “can generates” has been revised “can generate”; line 45: “originated from” has been revised “originating from”; line 91: “Stannous” has been corrected as “stannous”.

Round 2

Reviewer 1 Report

The authors did not answer all the questions.

There is still a space before the parenthesis between the poly and the rest of the polymer name.

“poly(butylene succinate-co-terephthalate)” – still o need to be in italic.

No answer on: “Due to the fact that there is a lot of research in this area (use of pentaerythritol as branching agent), please provide a clear aim and explicit novelty of the work”

I would like to ask the authors to carefully read the text and correct errors as well as to highlight the novelties and explain them in more detail.

Author Response

We have carefully considered the comments and revised our manuscript accordingly. Our point-to-point responses to the reviewers’ comments are also listed in the following response part.

There is still a space before the parenthesis between the poly and the rest of the polymer name.

“poly(butylene succinate-co-terephthalate)” – still o need to be in italic.

Response:We have gone through the full article and made corrections. Such as, poly (butylene dodecanedioate) has been revised poly(butylene dodecanedioate); poly(butylene succinate-co-terephthalate) has been corrected poly(butylene succinate-co-terephthalate). The similar error has been corrected in reference 19 and 25.

No answer on: “Due to the fact that there is a lot of research in this area (use of pentaerythritol as branching agent), please provide a clear aim and explicit novelty of the work”

I would like to ask the authors to carefully read the text and correct errors as well as to highlight the novelties and explain them in more detail.

Response:In fact, this branching work is very novel and the results are very interesting and useful. I believe it will attract and benefit researcher from many industrialized and fields. First, PBD is a very new type of biodegradable and biobased polyester. It is the first time to report the synthesis, structural characterization and branching PBD. Using biodegradable polymers is one of the effective ways to solve the problem of plastic pollution. However, most existing degradable polymers are composed of aliphatic linear units, so their relatively low mechanical properties, melt strength and heat resistance cannot satisfy the application’s need. Therefore, pentaerythritol is introduced into the macromolecular chains of PBD as a branching agent to improve the melt strength and mechanical properties of PBD. Second, film blowing is a great challenge for most biodegradable polyesters. This work is just to provide an effective strategy to resolve it. PBD with relatively high branching degree possesses good crystallinity and enhanced melt strength, which is beneficial to film blowing, and has potential applications in sustainable packaging.

Reviewer 3 Report

Authors significantly improved the manuscript answering all my questions and doubts. I suggest publication of the manuscript in this form.

Author Response

We really appreciate the reviewers’ professional comments and suggestions.

Reviewer 4 Report

I agree with the authors regarding the unreasonable turn-around time to submit a revision.  It is preposterous to assume that the authors are able to address major revisions in a couple days. 

Without adequate time, the authors are unable to address the review comments thoroughly.  However, without the recommended revisions, the recommendation would be to reject this manuscript as further commentary on the molecular details of the PBD are necessary for publication.

Author Response

Response:Thank you for your work on our manuscript. Your suggestion is very useful to improve the quality of our article.

In fact, we have systematically characterized the molecular structure; please see lines 120-127 (2.3. Characterization) and lines 192-223 (3.1 The chemical structures and compositions)

First, 1H NMR has been applied to characterize the molecular structure of the polymers, and the results testify that PBD and PBDx are our target products. It could be found from the partially enlarged spectra of Figure 2 (b) that a triplet appeared at 3.68 ppm, which attributed at methylene hydrogen near terminal hydroxyl group of the PBD and PBDx. The increased intensity of the 3.68 ppm with increasing PER content suggests more terminal hydroxyl group lies in the macromolecules.

In order to further confirm the molecular structure of the polymers, the chemical bonds in the polymers were analyzed by infrared spectroscopy, which showed that the absorption peaks at 1723 and 1266 cm-1, arising from C=O and C-O bonds on the ester bond, respectively. The intensity of the C-H, C=O and C-O bond absorption peaks increased with increasing PER concentration, indicating that the PER molecule has been successfully introduced into the PBD backbone.

Based on the above data, we believe the present characterization of molecular structure of the polymers is enough. Please tell us the detailed characterization of molecular details of the PBD needed to conduct further.

Round 3

Reviewer 1 Report

Thank you for explanation. I think a better explanation of the novelty should be included to the text.

Author Response

Thank you for explanation. I think a better explanation of the novelty should be included to the text.

Response:Thank you for your work on our manuscript. Your suggestion is very useful to improve the quality of our article. The better explanation of novelty has been provided in the revised article. Please see lines 76 to 78 and lines 88 to 95 on page 2, and lines 100-102 on Page 3 or the following:

Branching, as a very useful strategy to improve the mechanical and processing properties of linear polymers, attract and benefit researchers from many industrialized fields.

PBD is a very new type of biodegradable and biobased polyester and it is the first time to systematically study the synthesis, structural characterization of linear PBD and branched PBD. Development and application of biodegradable polymers is one of effective ways to solve the problem of plastic pollution. Nevertheless, biodegradable polymers are commonly based on aliphatic linear units with relatively low mechanical properties, melt strength and heat resistance, which cannot meet the needs of applications. In this work, PER was used as a branching agent to synthesize branched PBD to improve mechanical and rheological properties of PBD. Macromolecular chain structures of branched PBD were characterized, and the effect of branching on glass transition, mechanical properties and rheological properties was investigated. It could be found that branching resulted in improved melt strength, impact strength, tensile strength and tensile modulus. Second, film blowing is a great challenge for most biodegradable polyesters.

This work is just to provide an effective strategy to resolve it. PBD with relatively high branching degree possesses good crystallinity and enhanced melt strength, which is beneficial to film blowing, and the resulting branched PBD has potential applications in sustainable packaging.

Reviewer 4 Report

The manuscript is approved for publication.

Author Response

The manuscript is approved for publication.

Response: Thank you very much for your careful and thoughtful work on our manuscript ID Manuscript polymers-2478372 entitled “Long chain branched bio-based poly (butylene dodecanedioate) copolyester using pentaerythritol as branching agent: Synthesis, thermos-mechanical and rheological properties”.

Thanks again for your professional comments and suggestions, which is very helpful to improve the quality of our manuscript.
